

# A trace fossil made by a walking crayfish or crayfish-like arthropod from the Lower Jurassic Moenave Formation of southwestern Utah, USA

Makae Rose[1], Jerald D. Harris[2] and Andrew R.C. Milner[3]

[1] Department of Biology, Dixie State University, St. George, UT, USA
[2] Department of Physical Sciences, Dixie State University, St. George, UT, USA
[3] St. George Dinosaur Discovery Site at Johnson Farm, St. George, UT, USA

## ABSTRACT

New invertebrate trace fossils from the Lower Jurassic Moenave Formation at the St. George Dinosaur Discovery Site at Johnson Farm (SGDS) continue to expand the ichnofauna at the site. A previously unstudied arthropod locomotory trace, SGDS 1290, comprises two widely spaced, thick, gently undulating paramedial impressions flanked externally by small, tapered to elongate tracks with a staggered to alternating arrangement. The specimen is not a variant of any existing ichnospecies, but bears a striking resemblance to modern, experimentally generated crayfish walking traces, suggesting a crayfish or crayfish-like maker for the fossil. Because of its uniqueness, we place it in a new ichnospecies, *Siskemia eurypyge*. It is the first fossil crayfish or crayfish-like locomotion trace ever recorded.

## INTRODUCTION

Paleoichnology, the study of ichnofossils (fossil tracks and traces), contributes a substantial body of paleobiological information to the understanding of extinct organisms. This is because ichnofossils are direct results of ancient animal behavior (*Osgood, 1975*) that could only otherwise be inferred from body fossils. Furthermore, the global commonness of ichnofossils compared to body fossils means that the ichnological record often can preserve evidence of the presence of organisms not otherwise or poorly represented in the body fossil record (*Osgood, 1975*), especially of invertebrates that lack hard parts and therefore fossilize only under exceptional conditions. Except for conchostracans (sensu *Kozur & Weems, 2010*) and ostracods, which have biomineralized carapaces, arthropods, when compared to their evolutionary diversity, are among the less commonly preserved body-fossil components of terrestrial (including freshwater) paleoecosystems except in various Konservat-Lagerstätten (fossiliferous sites of exceptional preservational quality) (*Charbonnier et al., 2010*; *Luque et al., 2019*; *Selden & Nudds, 2012*; *Smith, 2012*). Yet from the mid-Paleozoic through the Cenozoic, arthropods—especially insects and arachnids— were certainly the most populous and diverse metazoan components of most terrestrial

Corresponding author
Jerald D. Harris, jharris@dixie.edu

ecosystems (*Labandeira & Beall, 1990*), and their paleoecological importance cannot be underestimated.

Arthropod ichnofossils can be more common and abundant than arthropod body fossils, and may indicate the presences of various arthropod taxa in terrestrial paleoecosystems for which body fossils may be entirely absent. Burrows (domichnia) made by arthropods comprise one such body of evidence. For example, several burrow ichnotaxa in eolian sandstones have been attributed to arthropods (*Ekdale, Bromley & Loope, 2007*). Some *Macanopsis*, *Psilonichnus*, and *Skolithos* burrows may have been made by spiders (*Uchman, Vrenozi & Muceku, 2018*); other *Psilonichnus* have been attributed to crabs (*Curran, Savarese & Glumac, 2016*). *Camborygma* burrows, attributed to crayfish (*Hasiotis & Mitchell, 1993*), are the primary body of evidence for crayfish in the fossil record. Perhaps more familiarly, walking tracks (repichnia) of arthropods have an extensive geological history, spanning from the Cambrian (and possibly latest Precambrian (*Chen et al., 2018*)) through the Holocene (*Eiseman & Charney, 2010*). They constitute some of the earliest evidence of metazoan life venturing onto land (reviewed in *Minter et al. (2016a, 2016b)*) and are known from virtually every paleoenvironment, from near shore and shallow marine environments (*Collette, Hagadorn & Lacelle, 2010*; *MacNaughton et al., 2002*; *Pirrie, Feldmann & Buatois, 2004*; *Shillito & Davies, 2018*; *Trewin & McNamara, 1994*) and, terrestrially, from proglacial systems (*Anderson, 1981*; *Lima, Minter & Netto, 2017*; *Lima et al., 2015*; *Uchman, Kazakauskas & Gaigalas, 2009*; *Walter, 1985*) to desert ergs (*Gilmore, 1927*; *Good & Ekdale, 2014*; *Sadler, 1993*).

The St. George Dinosaur Discovery Site at Johnson Farm (SGDS) in St. George, Washington County, Utah (Fig. 1) has been called a Konzentrat-Ichnolagerstätte (*Hunt & Lucas, 2006a*) because of its exceptional concentration of well-preserved ichnofossils from a broad spectrum of terrestrial organisms. The site preserves a detailed "snapshot" of an earliest Jurassic ecosystem from on- and offshore portions of a freshwater, lacustrine paleoenvironment. The "Dinosaur Discovery" part of the name of the site indicates the concentration of dinosaur tracks at this locality (*Milner, Lockley & Johnson, 2006*; *Milner, Lockley & Kirkland, 2006*; *Milner et al., 2009*), but tracks of other vertebrates (*Lockley, Kirkland & Milner, 2004*; *Milner, Lockley & Johnson, 2006*) and a moderately diverse invertebrate ichnofauna (*Lucas et al., 2006a*) are also preserved. Burrows pertaining to *Helminthoidichnites*, *Palaeophycus* and *Skolithos* are abundant at the site, but locomotory trackways made by arthropods, referred to cf. *Bifurculapes*, *Diplichnites* and cf. *Kouphichnium*, are also present. Ichnospecies of *Bifurculapes* have been variably attributed to insects, possibly beetles, and to crustaceans (*Getty, 2016*, *2018*; *Hitchcock, 1858*, *1865*); ichnospecies of *Diplichnites* have been attributed to myriapods (*Briggs, Rolfe & Brannan, 1979*; *Davis, Minter & Braddy, 2007*; *Pollard, Selden & Watts, 2008*; *Shillito & Davies, 2018*), notostracans (*Lucas et al., 2006a*; *Minter et al., 2007*), and other arthropods (*Melchor & Cardonatto, 2014*; *Minter et al., 2007*); and ichnospecies of *Kouphichnium* have been attributed to limulids (*Caster, 1944*; *King, Stimson & Lucas, 2019*; *Lomax & Racay, 2012*).

A previously unstudied SGDS specimen, SGDS 1290, is an arthropod locomotory trace that differs markedly from any other SGDS specimen, indicating the presence of a

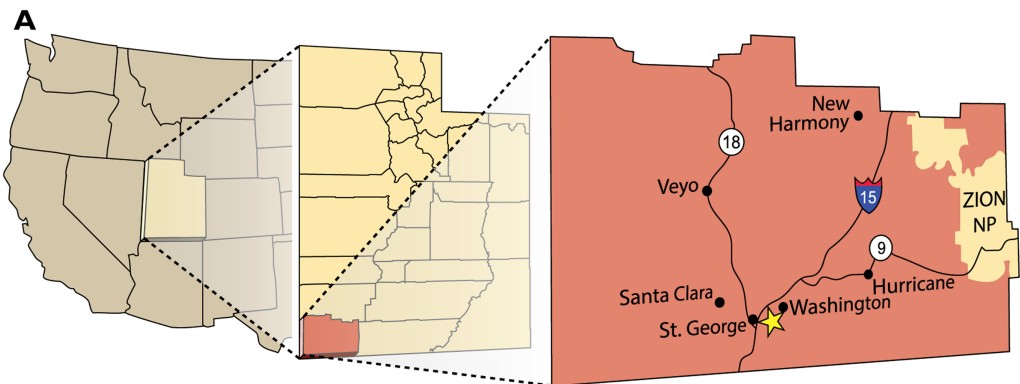

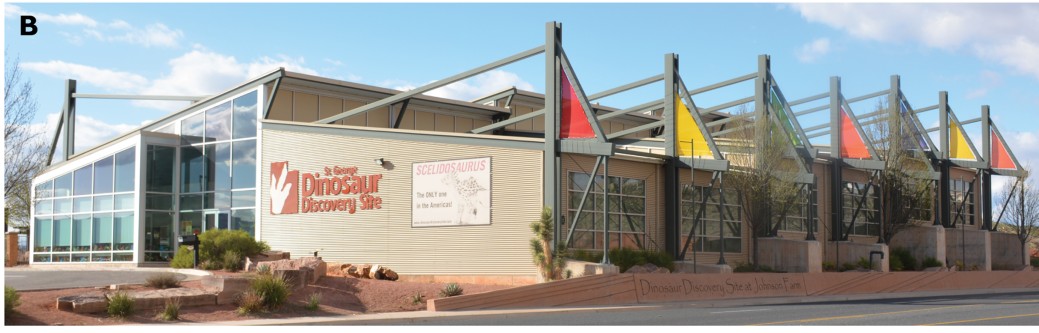

**Figure 1 Location of the St. George Dinosaur Discovery Site at Johnson Farm.** (A) Map showing the location of the St. George Dinosaur Discovery Site in St. George, Utah. (B) The museum at the St. George Dinosaur Discovery Site. Photograph by Jerald D. Harris. Figure modified from *Harris & Milner (2015)*; reproduced with permission from The University of Utah Press.

heretofore unrecognized component of the SGDS ichnofauna. SGDS 1290 is an arthropod locomotory trace because it includes distinct footprints in a discernible cycle, but lacks any features of vertebrate tracks, such as distinct toes (sensu *Seilacher, 2007*). The trace thus resembles numerous other fossil traces attributed to arthropods, as well as those generated experimentally. The fossil was discovered and collected 11 March 2010 by SGDS volunteer Jon Cross.

## GEOLOGICAL SETTING

Most of the fossils preserved at the SGDS, including the ichnofossil described below, occur in the Whitmore Point Member of the Moenave Formation (*Kirkland & Milner, 2006*; *Kirkland et al., 2014*), which conformably overlies the Dinosaur Canyon Member of the Moenave Formation and disconformably underlies the Springdale Sandstone Member, which itself has been assigned as both the lowest member of the Kayenta Formation (*Lucas & Tanner, 2006*) and the uppermost member of the Moenave Formation (*Steiner, 2014a*). The richest source of the ichnofossils at the SGDS, again including the trace described below, occurs within a fine-grained sandstone near the base of the Whitmore Point Member initially called the "Main Track Layer" (*Kirkland & Milner, 2006*; *Milner, Lockley & Johnson, 2006*; *Milner, Lockley & Kirkland, 2006*) and, later and more formally,

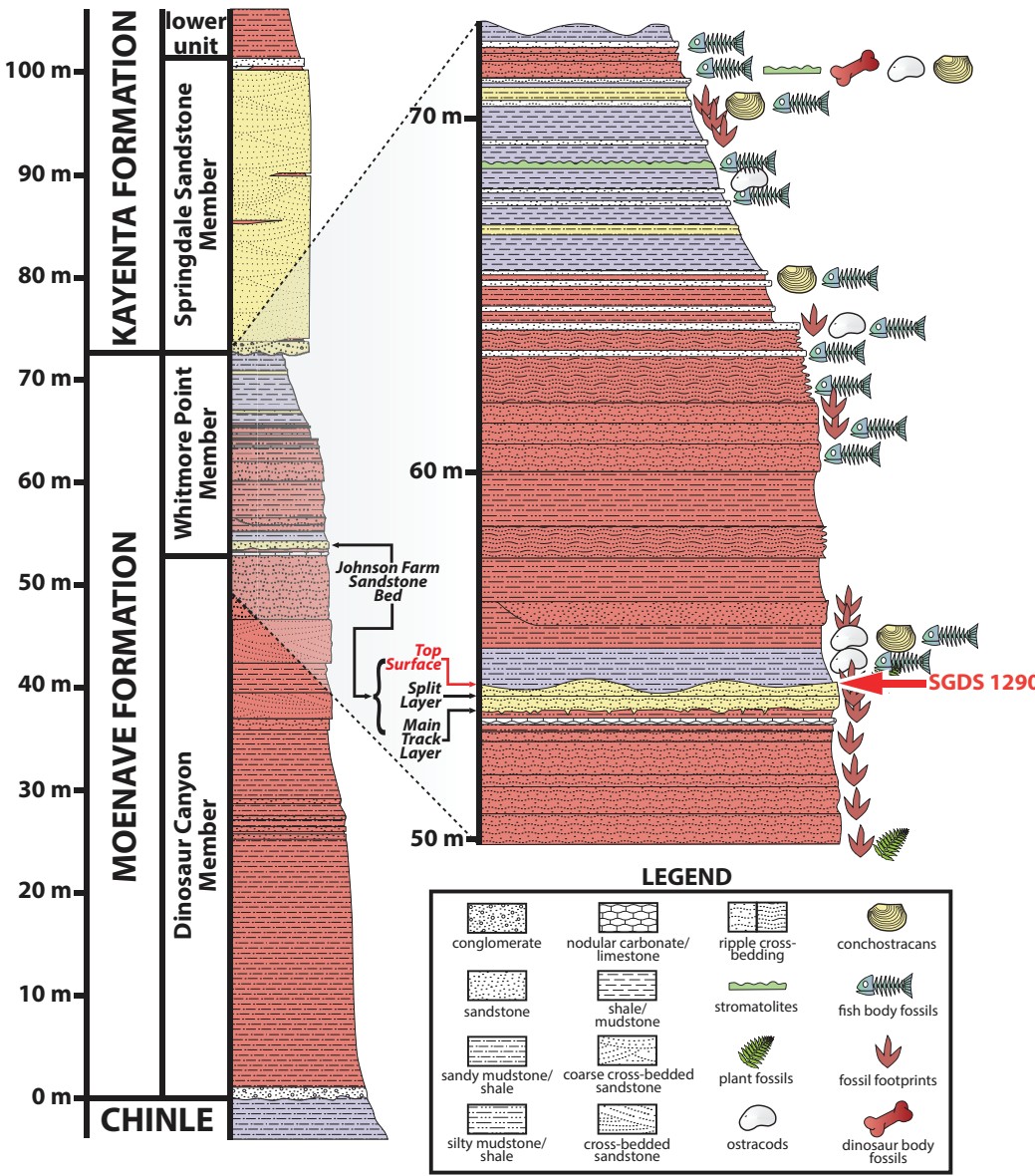

**Figure 2 Stratigraphic section at and immediately around the St. George Dinosaur Discovery Site (SGDS) in St. George, Utah.** Arthropod locomotory trackway SGDS 1290 comes from the "Top Surface Tracksite" layers of the Johnson Farm Sandstone Bed (red arrows). Figure modified from *Harris & Milner (2015)*; reproduced with permission from The University of Utah Press.

the Johnson Farm Sandstone Bed (unit 40 of *Kirkland et al., 2014*) (Fig. 2). SGDS 1290 comes from the uppermost strata of this unit, specifically one of several thinly bedded, apparently conformable, fine-grained-sandstone "Top Surface Tracksite" horizons (sensu *Kirkland et al., 2014*; *Milner, Lockley & Johnson, 2006*; *Milner, Lockley & Kirkland, 2006*). The Whitmore Point Member preserves sediments deposited in and around the large, freshwater Lake Whitmore (formerly called Lake Dixie) (*Kirkland & Milner, 2006*, *2014*); at the SGDS itself, the Johnson Farm Sandstone Bed preserves ichnofossils and sedimentary structures made in both subaerial and subaqueous conditions, indicating

a shoreline paleoenvironment (*Milner, Lockley & Kirkland, 2006*). Ichnologically, invertebrate ichnofossils in this paleoenvironment pertain to the *Scoyenia* ichnofacies (*Buatois & Mángano, 2004*; *Lucas et al., 2006a*), while the associated vertebrate ichnofauna pertains to the *Eubrontes* ichnocoenosis of the *Grallator* ichnofacies (*Hunt & Lucas, 2006b*, *2006c*).

Age determinations for the Whitmore Point Member have varied. The unit typically has been placed entirely within the Hettangian (earliest Jurassic) largely on biostratigraphic grounds (see discussions in *Kirkland et al. (2014)*, *Milner et al. (2012)*, *Parrish et al. (2019)* and *Tanner & Lucas (2009)*), but was also determined to straddle the Triassic–Jurassic boundary (201.3 ± 0.2 Ma) on magnetostratigraphic grounds (*Donohoo-Hurley, Geissman & Lucas, 2010*), in which system the Johnson Farm Sandstone Bed would be Rhaetian (latest Triassic) in age. However, *Steiner (2014b)* recovered Hettangian paleomagnetic sequences from the Whitmore Point Member, and *Suarez et al. (2017)* calibrated the magnetostratigraphic data of *Donohoo-Hurley, Geissman & Lucas (2010)* with high-precision U–Pb dates to re-situate the Triassic–Jurassic boundary stratigraphically farther down in the Dinosaur Canyon Member of the Moenave Formation, also making the Whitmore Point Member entirely earliest Jurassic in age. The Johnson Farm Sandstone Bed and its fossils therefore are Hettangian in age, approximately 200 million years old.

## MATERIALS AND METHODS

Ichnological terminology for arthropod locomotory traces used herein follows *Minter, Braddy & Davis (2007)* and *Genise (2017)*. *Minter, Braddy & Davis (2007)* defined "tracks" as discrete marks made by locomotory appendages, "impressions" as continuous traces made by another portion of the anatomy of a trace maker, and "imprints" as discontinuous such traces; they also provided terms for trackway arrangement and measurements. *Genise (2017)* outlined various descriptive terms for individual track morphologies. Measurements of SGDS 1290 (Fig. 3C; Table 1) were taken using digital calipers. The measurements were: track length and width, internal and external widths between paramedial impressions, distances between tracks and adjacent paramedial impressions, and widths of left and right paramedial impressions. Measurements pertaining to the paramedial impressions were taken adjacent to individual tracks/ track sets.

The electronic version of this article in Portable Document Format will represent a published work according to the International Commission on Zoological Nomenclature (ICZN), and hence the new names contained in the electronic version are effectively published under that Code from the electronic edition alone. This published work and the nomenclatural acts it contains have been registered in ZooBank, the online registration system for the ICZN. The ZooBank Life Science Identifiers can be resolved and the associated information viewed through any standard web browser by appending the LSID to the prefix http://zoobank.org/. The LSID for this publication is: urn:lsid: zoobank.org:pub:D78963CE-11C8-4447-8E26-BBCCF0E37143. The LSID for the herein described *Siskemia eurypyge* isp. nov. is: urn:lsid:zoobank.org:act:769B0815-8991-4F0E-

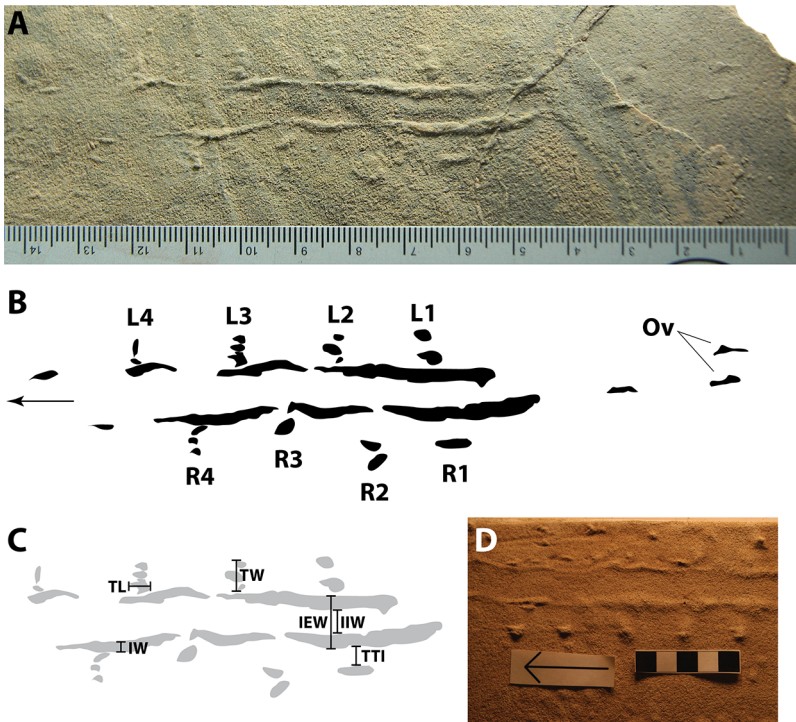

**Figure 3 Arthropod locomotory ichnofossil SGDS 1290.** (A) Photograph of specimen; scale in mm. (B) Schematic diagram of specimen. Arrow indicates direction of movement. L, left track; R, right track; Ov, overtracks. Numbers indicate position of tracks in sequence from posterior to anterior. Photograph by Andrew R.C. Milner. (C) Schematic diagram of specimen, showing examples of how measurements were taken. IEW, external width between paramedial impressions; IIW, internal width between paramedial impressions; IW, paramedial impression width; TL, track length; TTI, distance between track and adjacent paramedial impression; TW, track width. (D) Photograph of extant crayfish walking trace made in saturated, very fine sand for comparison to fossil in (A); scale in cm. Arrow indicates direction of travel; photograph rotated to have the same orientation as (A). Modified from *Fairchild & Hasiotis (2011*: fig. 4E*)*; reproduced with permission from SEPM.

**Table 1 Measurements (in mm) of arthropod locomotory trace fossil SGDS 1290.**

| Track Position | Length | Width | Impression internal width | Impression external width | Track to impression distance | Left impression width | Right impression width |
|---|---|---|---|---|---|---|---|
| L1 | 4.3 | 6.4 | 4.7 | 9.1 | 1.8 | 3.1 | 2.5 |
| L2 | 4.2 | 5.7 | 5.9 | 9.5 | 1.5 | 2.6 | 2.2 |
| L3 | 5.2 | 5.8 | 5.8 | 9.7 | 0.0 | 1.6 | 1.6 |
| L4 | 3.2 | 5.6 | (8.1) | (11.2) | 0.0 | 1.9 | 0.0 |
| R1 | 6.8 | 2.5 | 4.2 | 9.2 | 3.0 | 3.0 | 2.4 |
| R2 | 4.6 | 5.3 | 5.4 | 8.8 | 4.2 | 2.7 | 0.0 |
| R3 | 4.5 | 6.4 | 6.5 | 9.2 | 2.4 | 1.3 | 1.3 |
| R4 | 3.0 | 5.8 | 6.9 | 10.6 | 0.0 | 0.0 | 3.0 |

**Note:**
Measurements in parentheses are approximated based on faint portions of paramedial impressions.

B32C-99C87A9D293B. The online version of this work is archived and available from the following digital repositories: PeerJ, PubMed Central, and CLOCKSS.

## Description of SGDS 1290

SGDS 1290, a natural cast (convex hyporelief), consists of two parallel, undulating, paramedial ridges flanked externally by eight sets of small tracks that range in shape from ovoid to tapered (teardrop-shaped) to elongate (Figs. 3A and 3B). Tracks within each set are closely appressed; track sets are, however, spaced well apart from one another. The track sets are oriented perpendicular to the trackway axis, though tapered and elongate individual tracks within each set have long axes that parallel or are oblique to the trackway axis. Track sets average 4.5 mm long anteroposteriorly and 5.4 mm wide mediolaterally (Table 1). Left (L) and right (R) sets of tracks are arranged in an alternating pattern. Based on *Fairchild & Hasiotis (2011)*, the tapering ends of the tapered tracks are presumed to be anterior reflectures, indicating the direction of movement. Most tracks have long axes oriented parallel to the trackway axis; a few (such as in sets L1, L4 and R2) are oblique to the axis. Track R1 is markedly elongate rather than tapered, but also parallel to the trackway axis. Track sets L1, L3 and R4 consist of three distinct but appressed tracks; sets L2, L4 and R2 consist of pairs of appressed tracks, and R1 and R3 appear to consist of single tracks, although the possibility that each comprises multiple, conjoined tracks cannot be ruled out.

The paramedial impressions typically are thick mediolaterally, though they vary and taper briefly to nothingness in a few places (being more continuous than repeating, we consider them "impressions" and not "imprints"). The impressions follow gently undulating (non-linear and low amplitude) pathways. They span approximately 7.5 cm along the slab of rock. Overprints of short segments of the paramedial impressions that are not accompanied by tracks are visible behind the main trace segment on a slightly higher stratum. The impressions vary in width along their lengths, ranging from 0 to 3.1 mm (mean = 2.0 mm) for the left impressions and 0–3.0 mm (mean = 1.6 mm) for the right (Table 1). The width of the trace from left impression to right impression averages 9.4 mm when measured from the lateral (external) edges and 5.6 mm when measured between the medial (internal) edges (Table 1). The distances between the impressions thus are greater than the distances between the impressions and their flanking tracks (mean = 1.6 mm); the ratio of the distance between a paramedial impression and its flanking track to the distance between the medial edges of the paramedial impressions ranges from 0 to 0.78 (mean = 0.31; see Supplemental Material), so on average, the paramedial impressions are roughly three times farther apart from each other than either is from its flanking tracks. The impressions taper slightly in cross sectional view: they are wider at their bases and narrower at their rounded apices.

## Comparisons to Arthropod Repichnial Ichnotaxa

### Arthropod repichnia lacking medial or paramedial impressions

Several arthropod locomotory (walking) ichnotaxa are readily distinguished from SGDS 1290 by (usually) lacking medial or paramedial impressions, but are worth comparing to

SGDS 1290 to ascertain whether or not it might be a morphological variant of such ichnotaxa. These ichnotaxa are *Acanthichnus* (*Hitchcock, 1858*), *Asaphoidichnus* (*Miller, 1880*), *Bifurculapes* (*Hitchcock, 1858*), *Coenobichnus* (*Walker, Holland & Gardiner, 2003*), *Copeza* (*Hitchcock, 1858*), *Danstairia congesta* (*Smith, 1909*), *Diplichnites* (*Dawson, 1873*), *Eisenachichnus* (*Kozur, 1981*), *Foersterichnus* (*Pirrie, Feldmann & Buatois, 2004*), *Hamipes* (*Hitchcock, 1858*), *Laterigradus* (*De Carvalho et al., 2016*), *Lithographus* (*Hitchcock, 1858*), *Hornburgichnium* (*Kozur, 1989*), *Maculichna* (*Anderson, 1975a*), *Merostomichnites* (*Packard, 1900*), *Mirandaichnium* (*Aceñolaza, 1978*), *Octopodichnus* (*Gilmore, 1927*), *Petalichnus* (*Miller, 1880*), *Pterichnus* (*Hitchcock, 1865*), *Tasmanadia* (*Chapman, 1929*) and *Umfolozia* (*Savage, 1971*). Most of these ichnotaxa further differ from SGDS 1290 in the shapes and configurations of their tracks. The comparisons below specify track-making taxa only when one or more have been proposed for the ichnotaxon.

*Acanthichnus* tracks (Fig. 4A), attributed to a chelicerate such as a solifugid, are oppositely arranged, short, elongate impressions in two (or four, per *Dalman & Lucas, 2015*) parallel rows; tracks either are parallel to or angle slightly away from the trackway axis (*Dalman & Lucas, 2015*; *Hitchcock, 1858*). This morphology and organization are both unlike those of SGDS 1290.

*Asaphoidichnus* tracks (Fig. 4B), attributed to trilobites, are elongate to crescentic, possess 3–4 crescentic branches at one end, and are oriented oblique to the trackway axis (*Miller, 1880*). They are far more complex in structure than the tracks of SGDS 1290.

*Bifurculapes* (Fig. 4C), attributed to an insect, possibly a beetle (*Getty, 2016*), comprises adjacent pairs (rarely triplets) of slightly staggered, elongate, straight to crescentic tracks that lie parallel or slightly oblique to the trackway axis, unlike the tracks of SGDS 1290. Tracks in each pair sometimes converge toward one end in *Bifurculapes*. This ichnotaxon typically does not possess paramedial impressions, but a specimen described by *Getty (2016*: fig. 1*)* possess two such traces, albeit faintly, that lie closer to the tracks than to the trackway axis, as in SGDS 1290. These impressions are far less pronounced than their associated tracks, unlike those of SGDS 1290. *Getty (2020)* ascertained that *Bifurculapes* traces were made subaqueously and may have been made by a terrestrial insect that would have left different tracks subaerially.

*Coenobichnus* tracks (Fig. 4D), attributed to a hermit crab, are thick, crescentic to ellipsoidal, roughly parallel and closely appressed to the trackway axis, and asymmetrical, with the left tracks larger than the right tracks (*Walker, Holland & Gardiner, 2003*), all of which differentiate this ichnotaxon from SGDS 1290.

*Copeza* (possibly a variant and synonym of *Lithographus* (*Lull, 1953*; *Rainforth, 2005*; Fig. 4E)) consists of triplets of roughly oppositely arranged, linear, elongate tracks in which the anteriormost lies roughly perpendicular to the trackway axis while the posteriormost pairs lie parallel or oblique to the trackway axis (*Lull, 1953*). This rare ichnotaxon is thus unlike SGDS 1290.

*Danstairia congesta* (Fig. 4F) comprises circular to crescentic tracks in closely appressed sets of up to six that are oriented oblique to the trackway axis; tracks often overlap to form V-shaped structures (*Walker, 1985*), unlike in SGDS 1290.

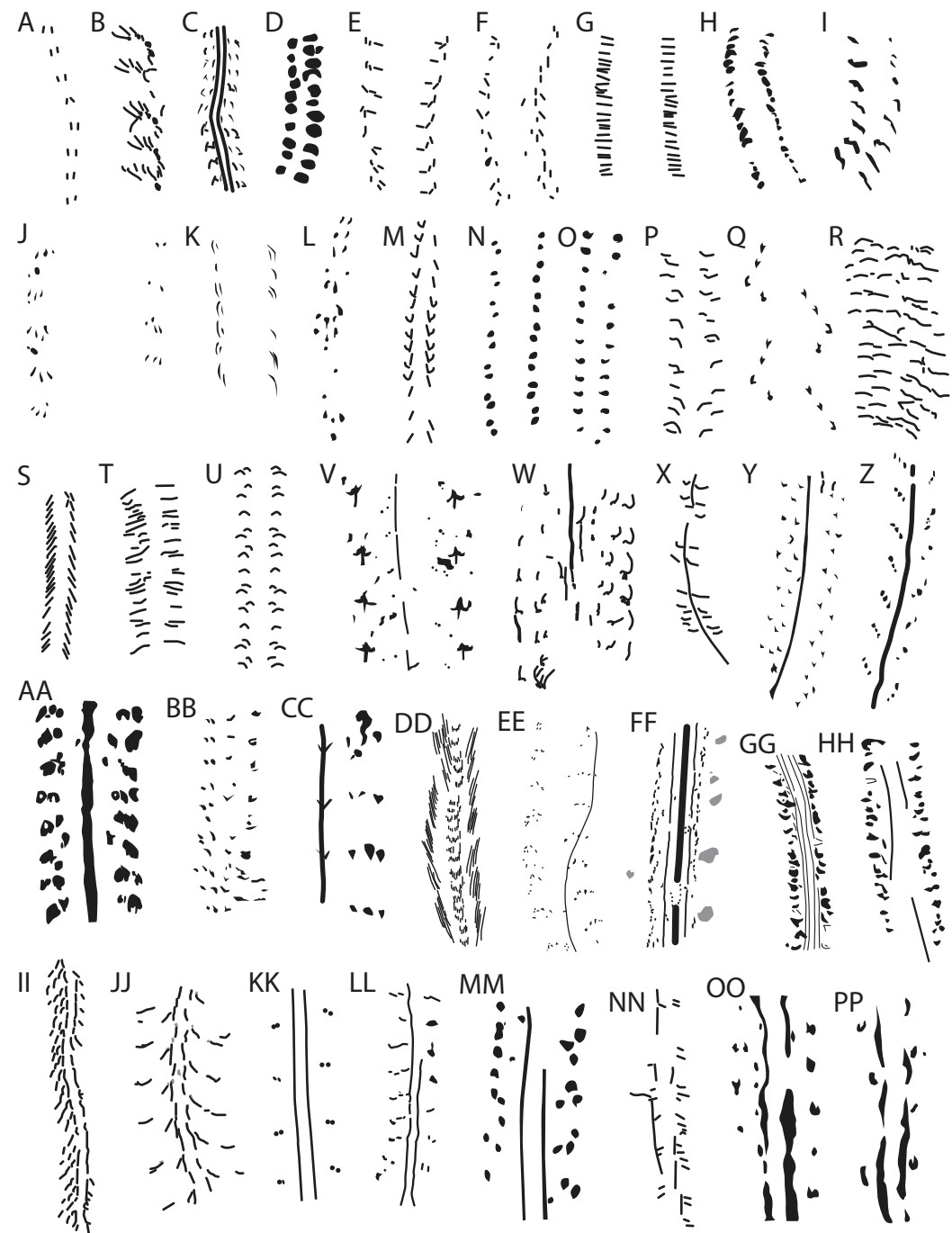

**Figure 4 Schematic diagrams of locomotory (presumably walking) ichnofossils attributed to arthropods (A–MM) and extant walking traces made by arthropods (NN–OO) compared to SGDS 1290 (PP).** Diagrams not to scale. (A) *Acanthichnus cursorius* (traced from *Hitchcock, 1858*). (B) *Asaphoidichnus trifidus* (traced from *Miller, 1880*). (C) *Bifurculapes laqueatus* (traced from *Getty, 2016*). (D) *Coenobichnus currani* (traced from *Walker, Holland & Gardiner, 2003*). (E) *Copeza triremis* (traced from *Hitchcock, 1858*). (F) *Danstairia congesta* (traced from *Walker, 1985*). (G) *Diplichnites aenigma* (traced from *Dawson, 1873*). (H) *Diplichnites gouldi* Type A (traced from *Trewin & McNamara, 1994*). (I) *Eisenachichnus inaequalis* (traced from *Kozur, 1981*). (J) *Foersterichnus rossensis* (traced from *Pirrie, Feldmann & Buatois, 2004*). (K) *Hamipes didactylus* (traced from *Getty, 2018*). (L) *Laterigradus lusitanicus* (traced from *De Carvalho et al., 2016*). (M) *Lithographus hieroglyphicus* (traced from

**Figure 4** (continued)
*Hitchcock, 1858*). (N) *Maculichna varia* (traced from *Anderson, 1975a*). (O) *Merostomichnites narrangansettensis* (traced from *Packard, 1900*). (P) *Mirandaichnium famatinense* (traced from *Aceñolaza, 1978*). (Q) *Octopodichnus didactylus* (traced from *Gilmore, 1927*). (R) *Petalichnus multipartatus* (traced from *Miller, 1880*). (S) *Pterichnus centipes* (traced from *Hitchcock, 1865*). (T) *Tasmanadia twelvetreesi* (traced from *Glaessner, 1957*). (U) *Umfolozia sinuosa* (traced from *Anderson, 1981*). (V) *Kouphichnium lithographicum* (traced from *Gaillard, 2011a*). (W) "*Merostomichnites* isp." (traced from *Hanken & Stormer, 1975*). (X) *Oniscoidichnus filiciformis* (traced from *Brady, 1947*). (Y) *Palmichnium antarcticum* (traced from *Braddy & Milner, 1998*). (Z) *Paleohelcura tridactyla* (traced from *Gilmore, 1926*). (AA) *Protichnites septemnotatus* (traced from *Owen, 1852*). (BB) *Robledoichnus lucasi* (traced from *Kozur & Lemone, 1995*). (CC) *Shalemichnus sittigi*, half of trackway (traced from *Kozur & Lemone, 1995*). (DD) *Stiallia berriana* (traced from *Smith, 1909*). (EE) *Stiaria quadripedia* (traced from *Walker, 1985*). (FF) *Mitchellichnus ferrydenensis* (traced from *Walker, 1985*). (GG) *Keircalia multipedia* (traced from *Walker, 1985*). (HH) *Danstairia vagusa* (traced from *Walker, 1985*). (II) *Glaciichnium liebegastensis* (traced from *Walter, 1985*). (JJ) *Warvichnium ulbrichi* (traced from *Walter, 1985*). (KK) *Siskemia bipediculus* (traced from *Walker, 1985*). (LL) *Siskemia elegans* (traced from *Walker, 1985*). (MM) *Siskemia latavia* (traced from *Walker, 1985*). (NN) Extant notostracan traces (traced from *Knecht et al., 2009*). (OO) Extant crayfish trace in saturated, very fine sand (traced from *Fairchild & Hasiotis, 2011*). (PP) SGDS 1290.

*Diplichnites* (possibly including *Acripes* per *Häntzschel (1975)* and *Hammersburg, Hasiotis & Robison (2018)*; also see below) tracks span a range of morphologies. *D. aenigma* (Fig. 4G), ostensibly the ichnospecies lectotype except that no specimen was designated as such (*Stimson et al., 2018*), typically comprises elongate, closely packed tracks in parallel rows on either side of the trackway axis; the tracks lie perpendicular to the trackway axis (*Dawson, 1873*). *D. gouldi* Type A (Fig. 4H) comprises parallel rows of closely spaced, oppositely arranged, simple, roughly circular to oblong to comma-shaped or irregular tracks with varying orientations to the trackway axis; *D. gouldi* Type B comprises closely spaced, elongate impressions oriented perpendicular, or nearly so, to the trackway axis, matching the general description of *D. aenigma*; *D. gouldi* Type C is similar to Type B, but the tracks are oriented oblique (~ 45°) to the trackway axis (*Trewin & McNamara, 1994*). *D. cuithensis*, attributed to large arthropleurid myriapods, is similar to both *D. aenigma* and *D. gouldi* Type B, but is very large and has widely spaced rows of tracks (*Briggs, Rolfe & Brannan, 1979*). *D. binatus* tracks often occur as closely appressed pairs of imprints (*Webby, 1983*). *D. triassicus* tracks are small and circular to ovoid rather than elongate, and frequently paired on either side of the trackway axis (*Pollard, Selden & Watts, 2008*); this ichnospecies has been alternately suggested to be a synonym of *D. gouldi* (*Lucas et al., 2006b*) or to pertain to *Acripes* (*Machalski & Machalska, 1994*; *Pollard, 1985*). *D. metzi* possesses a midline impression that is sometimes interrupted by connections between tracks in the closely appressed rows (*Fillmore et al., 2017*). The ichnogenus is in need of thorough review (*Smith et al., 2003*), but in all cases, the tracks are of different morphologies and arrangements than those of SGDS 1290.

*Eisenachichnus* tracks (Fig. 4I) are elongate, paired, and oblique to the trackway axis (rarely perpendicular), but the patterns of the pairs on either side of the trackway axis normally are asymmetrical: on one side, the paired tracks lie end to end, while on the other, they are more adjacent (*Kozur, 1981*). This morphology and arrangement are unlike those of SGDS 1290.

*Foersterichnus* (Fig. 4J), attributed to a crab, consists of widely spaced, paired rows of elongate tracks in sets of 3–4; rows are parallel to oblique to the trackway axis, and tracks in each set frequently overlap (*Pirrie, Feldmann & Buatois, 2004*). The wide spacing, clustering of tracks, and orientations of the tracks are unlike those of SGDS 1290.

*Hamipes* tracks (Fig. 4K) consist of closely spaced, paired, elongate to crescentic impressions oriented parallel to the trackway axis; the outer tracks are longer than their accompanying inner tracks, and the tracks are staggered or alternately arranged (*Getty, 2018*; *Hitchcock, 1858*). Track morphology readily differentiates *Hamipes* from SGDS 1290.

*Laterigradus* tracks (Fig. 4L), attributed to sideways-walking crabs, comprises asymmetrical trackways consisting of sets of up to four tracks (*De Carvalho et al., 2016*). Individual tracks vary widely in shape, ranging from elongate to tapered to comma-shaped to roughly circular. Track sets fall within a narrow trackway width and exhibit different stride lengths along the course of a trackway. While some individual track shapes resemble those of SGDS 1290, the overall arrangement and layout of the tracks is distinct.

*Lithographus* (including *Permichnium* sensu *Minter & Braddy, 2009*) tracks (Fig. 4M), which match those made by pterygote insects, especially extant cockroaches (*Davis, Minter & Braddy, 2007*), comprise trios (or pairs, in the case of the *Permichnium* variant) of elongate to comma-shaped, rather than circular or tapering, tracks that are arranged at different angles to one another, some of which are oriented perpendicular to the trackway axis, and others of which are oblique to the trackway axis (*Guthörl, 1934*; *Hitchcock, 1858*; *Minter & Braddy, 2009*). These track morphologies and arrangements are distinct from those of SGDS 1290. *Hornburgichnium* reportedly is similar to *Permichnium*, but has three tracks on either side of the midline instead of two, and at least one of each set is oriented parallel to the trackway axis (*Kozur, 1989*); it may also be a variant of *Lithographus* (*Lucas et al., 2005b*). Trackways of *Lithographus* can transition into trackways that *Hitchcock (1858)* called *Hexapodichnus* (*Davis, Minter & Braddy, 2007*; *Minter, Braddy & Davis, 2007*), so the latter may be considered a behavioral and/or substrate-consistency variant of the former, and also unlike SGDS 1290.

*Maculichna* (including *Guandacolichnus* and *Paganzichnus* of *Pazos (2000)* per *Buatois & Mángano (2003)*) tracks (Fig. 4N) comprise pairs (sometimes more) of small, circular to slightly elongate tracks arranged in closely appressed, slightly staggered rows. Pairings are oriented virtually parallel to the trackway axis (*Anderson, 1975a*); occasionally, short segments of linear, singular medial or closely spaced, paired paramedial imprints are also preserved that can be offset to one side of the trackway axis (*Anderson, 1975a*: figs. 8b, 8d and 8e). The pairing of *Maculichna* tracks differs from that of SGDS 1290. *Aceñolaza & Buatois (1991, 1993)* and *Archer & Maples (1984)* described *Maculichna* traces that exhibit the pairing of classic *Maculichna* from South Africa, but in which tracks are more ellipsoidal to shaped like slightly inflated isosceles triangles; the long axes of the triangles are oriented close to perpendicular to the trackway axis. *Pazos (2000)* recognized this morphology as the separate ichnotaxon *Paganzichnus*. This morphology is also unlike that of SGDS 1290.

The ichnospecies holotype of *Merostomichnites narragansettensis* (Fig. 4O) consists of parallel rows of roughly oppositely arranged circular to elongate to comma-shaped tracks, the long axes of which are perpendicular to the trackway axis (*Packard, 1900*). *Merostomichnites beecheri* tracks are circular and connected across the trackway axis by curvilinear, shallow, M-shaped imprints, creating a sort of segmented midline impression (*Packard, 1900*: fig. 4). These track and trace morphologies do not match those of SGDS 1290.

*Mirandaichnium* (Fig. 4P) consists of two rows of elongate, linear tracks that terminate laterally in small, circular impressions. Tracks are oriented perpendicular or oblique to the trackway axis, oppositely situated, and often grouped into series of eight (*Aceñolaza & Buatois, 1993*; *Buatois et al., 1998*), unlike those of SGDS 1290.

*Octopodichnus* (Fig. 4Q) ichnospecies, attributed to arachnids, have different morphologies. *O. didactylus* tracks are circular to crescentic to bifurcate or trifurcate oriented parallel to the trackway axis in alternating, arcuate sets of four (*Sadler, 1993*). *O. minor* tracks have a similar organization, but the tracks are more amorphous; *O. raymondi* tracks consist of clusters of four circular to crescentic marks arranged in checkmark-like patterns (*Sadler, 1993*). These track morphologies and distinctive arrangements are substantially unlike those of SGDS 1290.

*Petalichnus* (Fig. 4R), attributed to trilobites (*Braddy & Almond, 1999*), comprises sets of 2–3 elongate to crescentic tracks oriented perpendicular to the trackway axis (*Miller, 1880*). *Anderson (1975b)* and *Braddy & Almond (1999)* diagnosed *Petalichnus* tracks as sometimes bifurcate and occurring in series of 9–12; they further noted that the ichnotaxon needs review. Both track morphology and organization are unlike those of SGDS 1290.

*Pterichnus* tracks (Fig. 4S), attributed to isopods, frequently are segmented and are more linear and elongate than any in SGDS 1290. Tracks (or series of segments) are oriented oblique (Types 1 and 2 of *Gaillard et al., 2005*) or parallel (Types 3 and 4 of *Gaillard et al., 2005*) to the trackway axis, and approximately symmetrically arranged in two parallel rows (*Gaillard et al., 2005*; *Hitchcock, 1865*) that are somewhat closer together than are those of the morphologically similar *Diplichnites*. Types 3 and 4 of *Gaillard et al. (2005)* morphologically grade into *Diplopodichnus* (*Uchman et al., 2011*). *Hammersburg, Hasiotis & Robison (2018)* suggested that *Pterichnus* comprises undertracks of *Lithographus* and is thus a junior synonym of that ichnotaxon. In any case, *Pterichnus* tracks are readily distinguished from SGDS 1290.

*Tasmanadia* traces (Fig. 4T) consist of two rows of closely packed, elongate, linear tracks oriented generally perpendicular to the trackway axis; occasionally, tracks overlap at one end, creating narrow, V-shaped structures (*Chapman, 1929*; *Glaessner, 1957*). Morphologically, this ichnotaxon resembles *Diplichnites gouldi* Type B and *Umfolozia* (but lacks the organization of the latter), and differs from SGDS 1290 for the same reasons as those ichnotaxa.

*Umfolozia* (Fig. 4U), attributed to syncarid and peracarid crustaceans (*Lima, Minter & Netto, 2017*; *Savage, 1971*), consists of parallel rows of irregularly shaped to crescentic tracks oriented perpendicular or oblique to the trackway axis that follow a unique

repeating pattern (*Anderson, 1981*; *Savage, 1971*) unlike anything discernible in SGDS 1290. *Aceñolaza & Buatois (1993)* noted morphological similarities between *Mirandaichnium*, *Tasmanadia*, and *Umfolozia* and postulated similar track makers.

In summary, SGDS 1290 is not a variant of any of these ichnotaxa.

### Arthropod repichnia possessing one medial impression

Several other arthropod locomotory (walking) ichnotaxa are also readily distinguished from SGDS 1290 by possessing singular medial, rather than paired paramedial, impressions; again, comparison is warranted to ascertain whether or not SGDS 1290 is a morphological variant of such ichnotaxa. These ichnotaxa are *Kouphichnium* (*Caster, 1938*; *Nopcsa, 1923*), "*Merostomichnites* isp." (*Hanken & Stormer, 1975*), *Oniscoidichnus* (*Brady, 1947*, *1949*), *Palmichnium* (*Richter, 1954*), *Paleohelcura* (*Gilmore, 1926*), *Protichnites* (*Owen, 1852*), *Robledoichnus* (*Kozur & Lemone, 1995*), *Shalemichnus* (*Kozur & Lemone, 1995*), *Stiallia* (*Smith, 1909*) and *Stiaria* (*Smith, 1909*). As with traces lacking any medial impressions, these ichnotaxa further differ from SGDS 1290 in track morphology. As before, possible track makers are specified only when available.

*Kouphichnium* traces (Fig. 4V) are attributed to limulids and occur in a variety of configurations. Tracks in clear *Kouphichnium* walking traces that possess singular medial impressions (e.g., many *K. lithographicum*, but not, for example, *K. minkinensis* (*King, Stimson & Lucas, 2019*; q.v. *Gaillard, 2011a*; *Shu et al., 2018*)) typically occur in sets of up to five in rows oriented oblique to the medial impression and trackway axis; individual tracks range from circular and ellipsoidal to elongate, and can split into anywhere from 2 to 5 branches at their ends (*Caster, 1938*, *1944*; *King, Stimson & Lucas, 2019*; *Shu et al., 2018*). Well-preserved *Kouphichnium* tracks are dissimilar to those of SGDS 1290.

Traces referred to as "*Merostomichnites*" (Fig. 4W) and attributed to the eurypterid *Mixopterus* by *Hanken & Stormer (1975)* consist of three elongate and crescentic tracks in oblique rows on either side of an intermittent medial impression; the tracks increase in size laterally, and some split into two or more branches on one end. In any of these cases, however, the tracks are substantially more complex than those of SGDS 1290.

*Oniscoidichnus* tracks (Fig. 4X) are elongate to crescentic, oriented perpendicular or oblique to the trackway axis, closely packed and closely appressed to the single midline impression (*Brady, 1947*; *Davies, Sansom & Turner, 2006*). In all these details, *Oniscoidichnus* traces differ markedly from SGDS 1290.

Ichnospecies of *Palmichnium* (Fig. 4Y), also attributed to eurypterids, vary in morphology. Generally, they comprise complex sets of tracks lying lateral to a medial impression that can be either continuous or discontinuous. Tracks range in shape from elongate to crescentic to ovoid to chevron shaped, and they generally parallel the trackway axis. The tracks occur in oblique rows in sets of up to four; in some traces, the lateralmost tracks are elongate and curved, while the more medial tracks are linear and oriented parallel to the trackway axis (*Braddy & Milner, 1998*; *Minter & Braddy, 2009*; *Poschmann & Braddy, 2010*; *Richter, 1954*). Tracks are both more numerous and differently shaped than those of SGDS 1290.

*Paleohelcura* (including *Mesichnium* per *Braddy (1995)* and *Triavestigia* per *Kozur, Löffler & Sittig (1994)*; possibly a junior synonym of *Stiaria*; Fig. 4Z) traces, attributed to scorpions (*Brady, 1947*; *Davis, Minter & Braddy, 2007*), comprise small, circular tracks in sets of three in either rows, triangular arrangements, or checkmark-like patterns that lie external and oblique to the medial impression (*Gilmore, 1926*; *Lagnaoui et al., 2015*; *Sadler, 1993*). This distinctive layout is unlike that of SGDS 1290. *De Peixoto et al. (2020)* attributed traces lacking a medial impression and comprising closely appressed pairs or triplets of mostly elliptical tracks from the Upper Jurassic or Lower Cretaceous of Brazil to a new ichnospecies of *Paleohelcura* and attributed them to a pterygote insect track maker. Tracks in this ichnospecies are arranged in rows oriented oblique to the trackway axis, and track sets in this ichnospecies lie close to the midline. This morphology is also unlike that of SGDS 1290.

*Protichnites* traces (Fig. 4AA) comprise thick, often segmented medial impressions (sometimes absent except on trackway turns) flanked by oppositely arranged, subcircular to ellipsoidal to irregularly shaped tracks with varying orientations to the trackway axis (*Burton-Kelly & Erickson, 2010*; *Collette, Gass & Hagadorn, 2012*; *Hagadorn & Seilacher, 2009*). They differ substantially from the tracks of SGDS 1290.

*Robledoichnus* tracks (Fig. 4BB), attributed to flying insects, resemble tracks of *Eisenachichnus* but possess a discontinuous, faint medial trace consisting entirely of periodic, V-shaped marks flanked by asymmetrical pairs of tracks. On one side, the tracks are short, tapered, and oriented oblique to the trackway axis; on the other side, the tracks are longer and crescentic, oriented closer to perpendicular to the trackway axis (*Kozur & Lemone, 1995*). *Lucas et al. (2005a)* considered *Robledoichnus* a probable junior synonym of *Paleohelcura* or *Stiaria*, and the ichnotaxon differs from SGDS 1290 for similar reasons as those ichnotaxa, in addition to the asymmetry.

*Shalemichnus* traces (Fig. 4CC), for which only half a trackway is known, include a straight medial impression punctuated at intervals by V-shaped marks. This impression is flanked by sets of three tapered tracks in straight rows oriented perpendicular to the trackway axis; individual tracks have their long axes parallel to the trackway axis (*Kozur & Lemone, 1995*). *Minter & Braddy (2009)* considered *Shalemichnus* a junior synonym of *Stiaria*. The tracks of *Shalemichnus* bear some similarity to those of SGDS 1290, but the paramedial impressions of SGDS 1290 lack the V-shaped markings of the medial impression of *Shalemichnus*.

*Stiallia* traces (Fig. 4DD) consist of paired rows of long, linear impressions parallel or slightly oblique to the trackway axis and that frequently overlap. *Stiallia pilosa* lacks any medial or paramedial impressions, but *Stiallia* (*Carrickia* of *Smith (1909)*) *berriana* possesses a medial row of crescentic to chevron-shaped marks (*Smith, 1909*; *Walker, 1985*). *Pollard (1995)* suggested that *Stiallia* could be an arthropod swimming, rather than a walking, trace, though it also resembles traces made by bristletail insects walking in highly saturated mud (*Getty et al., 2013*: figs. 6F and 6G). *Stiallia* tracks are markedly unlike those of SGDS 1290.

*Stiaria* tracks (including some ichnospecies of *Danstairia* of *Smith (1909)*; Fig. 4EE), attributed to scorpionids (*Braddy, 2003*; *Lucas, Lerner & Voigt, 2013*) and monuran insects

(*Genise, 2017*; *Kopaska-Merkel & Buta, 2013*), are oppositely situated groups of 2–4 generally circular to tapered tracks in a linear to crescentic arrangement lying roughly perpendicular to the trackway axis (*Walker, 1985*). In some specimens and ichnospecies of *Stiaria*, the singular medial impression actually varies in position, meandering from medial to almost lateral to their tracks (*Fillmore, Lucas & Simpson, 2012*: fig. 26d; *Walker, 1985*: figs. 5b and 5c). In some Mississippian specimens from Pennsylvania, the medial impression is flanked by thin, discontinuous, but closely appressed paramedial imprints (*Fillmore, Lucas & Simpson, 2012*: figs. 26d–26g). Track arrangement alone differentiates *Stiaria* from SGDS 1290. *Genise (2017)* asserted that *Stiaria* should be considered a junior synonym of *Siskemia* (the latter has page priority over the former).

As with locomotory traces lacking medial impressions, SGDS 1290 is not a variant of any of these ichnotaxa.

### Arthropod repichnia possessing three or more medial and paramedial impressions

*Mitchellichnus* (Fig. 4FF), attributed to archaeognathan insects (*Getty et al., 2013*), is distinguished from SGDS 1290 by possessing three medial impressions (*Walker, 1985*). *Mitchellichnus* tracks are complex, comprising two distinct types and arrangements. An inner set, lying close to the medial impressions, consists of apparently elongate tracks in sets of up to six that lie parallel to slightly oblique to the trackway axis; an outer set consists of larger, amorphous impressions (*Walker, 1985*). Tracks are thus more numerous in *Mitchellichnus* than in SGDS 1290, and the tracks differ in arrangement and morphology. Like *Stiaria*, *Genise (2017)* asserted that *Mitchellichnus* should be considered a junior synonym of *Siskemia*.

*Keircalia* (Fig. 4GG) is distinguished from SGDS 1290 by possessing four medial impressions (*Smith, 1909*; *Walker, 1985*). *Keircalia* tracks are crescentic to irregularly shaped, generally are oriented perpendicular to the trackway axis, and have no discernible arrangement (*Walker, 1985*). Both track morphology and organization are unlike those of SGDS 1290.

### Arthropod repichnia possessing paired paramedial impressions

A few ichnotaxa, as well as some experimentally produced tracks of extant arthropods, resemble SGDS 1290 by possessing paired paramedial impressions in at least some specimens. Such ichnotaxa are *Danstairia vagusa* (*Smith, 1909*), *Glaciichnium* (*Walter, 1985*), *Warvichnium* (*Walter, 1985*), and *Siskemia* (*Smith, 1909*); similar extant traces include those made by notostracans (*Trusheim, 1931*) and crayfish (*Fairchild & Hasiotis, 2011*).

*Danstairia vagusa* (Fig. 4HH) possesses intermittent, thin, linear imprints that do not always parallel their accompanying trackways. Tracks are circular to triangular, generally have their long axes perpendicular to the trackway axis, and lack any coherent layout (*Walker, 1985*), unlike those of SGDS 1290. *D. vagusa* somewhat resembles *Keircalia* traces, but its tracks are spaced more widely apart.

*Glaciichnium* traces (Fig. 4II), which resemble traces made by isopods (*Gibbard & Stuart, 1974*; *Lima, Minter & Netto, 2017*; *Uchman, Kazakauskas & Gaigalas, 2009*; *Uchman et al., 2011*), comprise 1–3 elongate, linear tracks ("bars" that are divided into segments (*Uchman, Kazakauskas & Gaigalas, 2009*)) that lie oblique to the trackway axis and are staggered on either side of that axis, unlike the tracks of SGDS 1290; their linear, serial but discontinuous paramedial imprints are widely spaced, consistently abutting the medial ends of the tracks (*Walter, 1985*), farther apart than those of SGDS 1290. *Lima et al. (2015)* described the paramedial imprints in Brazilian specimens as comprising successive pairs of C-shaped imprints rather than strictly linear structures, further unlike SGDS 1290. Some *Glaciichnium* traces also possess a medial imprint as well (*Uchman, Kazakauskas & Gaigalas, 2009*; *Walter, 1985*).

*Warvichnium* traces (Fig. 4JJ) are complex, comprising pairs to multiple sets of linear, discontinuous medial and paramedial imprints flanked by varying numbers of linear to crescentic tracks in two or more sets: an inner set, close to the medial imprints, that are oriented slightly oblique to the trackway axis, and an outer set oriented closer to perpendicular to the trackway axis (*Walter, 1985*), quite unlike SGDS 1290. *Getty (2020)* noted similarities between *Warvichnium* and subaqueous *Bifurculapes*.

Among described arthropod repichnia, SGDS 1290 architecturally most closely resembles ichnospecies of *Siskemia* by possessing discreet, compact (not linear) tracks and track sets flanking paired paramedial impressions. Three ichnospecies of *Siskemia* are presently recognized (*Walker, 1985*):

- *S. bipediculus* (Fig. 4KK) comprises small, circular tracks in closely appressed pairs (occasionally trios) in rows oriented perpendicular or slightly oblique to the trackway axis; the pairs are spaced apart at approximately regular intervals and evenly distant from the uniformly straight and narrow paramedial impressions (*Walker, 1985*). The paramedial impressions lie close to the midline axis, well away from their adjacent tracks (the average ratio of the distance between a paramedial impression and its flanking track to the distance between the medial edges of the paramedial impressions is 1.34; see Supplemental Material).

- *S. elegans* (Fig. 4LL) has similarly shaped tracks in sets of up to four; the sets similarly lie well away from the likewise straight, narrow, and closely appressed paramedial impressions (the average ratio of the distance between a paramedial impression and its flanking track to the distance between the medial edges of the paramedial impressions is 1.75; see Supplemental Material). *S. bipediculus* and *S. elegans* differ primarily in the orientations of their track rows to the trackway axis and the continuities and thicknesses of their paramedial impressions (*Walker, 1985*), though these could be behavioral and/or substrate-driven variants.

- *Siskemia latavia* (Fig. 4MM; called "*lata-via*" by *Smith (1909)* and *Walker (1985)*, but the ICZN does not permit hyphens in genus or species names) tracks comprise tapered or ovoid tracks arranged in roughly triangular sets of three. Most individual tracks have their long axes oriented oblique to the trackway axis; track sets have varying orientations to the trackway axis. Tracks in individual *S. latavia* sets usually are spaced

well apart from each other; rarely do two individual tracks in a set contact one another. *S. latavia* tracks lack the regular spacing and arrangements of those of *S. elegans* and *S. bipediculus*, and occasional individual tracks lie close to the paramedial impressions, farther medially than in either of the other two *Siskemia* ichnospecies (the average ratio of the distance between a paramedial impression and its flanking track to the distance between the medial edges of the paramedial impressions is 0.80; see Supplemental Material). Each paramedial impression of *S. latavia* is slightly wider than those of the other two *Siskemia* ichnospecies (probably a function of the larger overall size of specimens attributed to this ichnospecies), but retain the close appression to the trackway midline and the uniform straightness.

The tracks of SGDS 1290 vary more in morphology than those of any known *Siskemia* ichnospecies, but grossly share their layout. Tracks in all three *Siskemia* ichnospecies have a staggered distribution, similar to, but less pronounced than, that of SGDS 1290. SGDS 1290 differs most markedly from any of the three *Siskemia* ichnospecies in the morphology and positions of the paramedial impressions: in SGDS 1290, the impressions vary in thickness along their lengths and undulate, in contrast to the thin, straight impressions of all three *Siskemia* ichnospecies. Additionally, the impressions in SGDS 1290 lie farther apart than those of the three *Siskemia* ichnospecies. In fact, all of the ichnospecies of *Siskemia* erected by *Smith (1909)*, as well as both specimens later attributed to this ichnogenus (*Getty et al., 2013*; *McNamara, 2014*; *Pollard, Steel & Undersrud, 1982*) and *Siskemia*-like traces made by extant, archaeognathan insects (*Getty et al., 2013*), have such thin, linear, closely appressed paramedial impressions (sometimes offset toward one side of the trackway). The only time when archaeognathan traces approach the paramedial impression spacing of SGDS 1290 is when both abdominal styli (laterally) and gonostyli (medially) of the trace makers register impressions and imprints in wet mud, producing two sets of paramedial traces (*Getty et al., 2013*: figs. 6K and 6L), but even then the linear, lateralmost paramedial impressions still do not resemble the thick, undulating impressions of SGDS 1290. Simultaneously, in such wet mud, archaeognathan tracks are elongate and oriented oblique to the trackway axis, unlike those in SGDS 1290. In total, SGDS 1290 does not fit neatly into any known *Siskemia* ichnospecies and does not seem to be an archaeognathan insect trace.

Among traces made by extant arthropods, SGDS 1290 bears similarities to traces made by both notostracans and crayfish. Interpretive drawings of experimental traces made by notostracans figured by *Trusheim (1931)* depict elongate, crescentic, or tapered tracks oriented perpendicular to paramedial impressions; the tracks are arranged oppositely, unlike those of SGDS 1290. Additionally, the thin, linear paramedial impressions figured by *Trusheim (1931)* lie so far from the trackway axis that they often contact their accompanying tracks, a phenomenon that only occurs in SGDS 1290 near L4 and R4, where the lateral margins of its undulating paramedial impressions meander particularly far laterally. *Tasch (1969)* noted, though, that the drawings presented by *Trusheim (1931)* were misleading compared to his own experimentally generated notostracan traces. However, he described the morphologies of his notostracan tracks only as "minute en

echelon stripes" (*Tasch, 1969*: 327), which does not adequately specify how they differed from those of *Trusheim (1931)*; track details are impossible to discern in his lone photographic figure (*Tasch, 1969*: pl. 1.2). *Gand et al. (2008)* also conducted neoichnological experiments with notostracans, recovering locomotory traces that were less orderly than those illustrated by *Trusheim (1931)* (*Gand et al., 2008*: figs. 16.1, 16.2 and 17.1). Their extant notostracan tracks comprised multiple tracks with rather chaotic distributions lateral to their paramedial impressions, unlike the regular distribution seen in SGDS 1290. *Gand et al. (2008)* found their notostracan traces to fall within the "etho-morphotype" of *Acripes*, as exemplified by their novel ichnospecies *A. multiformis* from the Permian of France. (*Linck (1943)* and *Pollard (1985)* also referred *Acripes* (*Merostomichnites* of *Linck (1943)*) tracks to notostracans, but not based on neoichnological experiments.) *A. multiformis* traces, unlike classic *Acripes* (*Matthew, 1910*), possess paramedial imprints, albeit inconsistently. *Hammersburg, Hasiotis & Robison (2018)*, *Häntzschel (1975)*, *Miller (1996)* and *Pemberton, MacEachern & Gingras (2007)* all supported classic *Acripes* as a junior synonym of *Diplichnites*; the issue of synonymy is beyond the scope of this paper, but we note at least that the tracks in fossils that *Gand et al. (2008)* called *A. multiformis* differ from SGDS 1290 in the same ways that *Diplichnites* tracks do (see above). Lastly, *Knecht et al. (2009*: figs. 5 and 6) also illustrated traces made by extant notostracans (Fig. 4NN), which are "tidier" than those of *Gand et al. (2008)* and resemble those of classic *Acripes* and *Diplichnites*, albeit with discontinuous paramedial and curvilinear lateral (external) imprints. The tracks in these traces comprise irregular, ellipsoidal sets oriented oblique to the trackway axis, unlike those of SGDS 1290. In total, SGDS 1290 is unlikely to be a notostracan trace.

*Fairchild & Hasiotis (2011)* conducted neoichnological experiments with crayfish to examine their locomotory traces. These traces varied in morphology depending on substrate conditions (sediment grain size and saturation) and slope; in general, when clearest, they consist of sets of 1–4 circular, tapering, ellipsoidal, or elongate tracks, occasionally of different sizes, that are oriented parallel to the trackway axis and that lie lateral to a pair of undulating, variably thick paramedial impressions that lie closer to their flanking tracks than to each other (Figs. 3D and 4OO). Morphologically, the tracks and impressions match those of SGDS 1290, although the tracks made by extant crayfish often are larger than those of SGDS 1290 when produced in dry substrate (*Fairchild & Hasiotis, 2011*: fig. 9). Track sets in extant crayfish traces have complex arrangements: when comprised of multiple traces, they frequently are in rows oriented perpendicular to the trackway axis, but sometimes rows are oblique to the trackway axis. When fewer tracks are preserved, sets can appear to lie in single, parallel rows on either side of the paramedial impressions. Track positions can be opposite to staggered to alternating, also as in SGDS 1290. In both track and paramedial impression morphology, as well as in overall trace architecture, SGDS 1290 strongly resembles crayfish traces made in damp silt and clay (*Fairchild & Hasiotis, 2011*: figs. 2e and 2f), dry and saturated, very fine-grained sand (Fig. 3D; *Fairchild & Hasiotis, 2011*: figs. 4a, 4b, 4e and 4f), dry and damp, fine-grained sand (*Fairchild & Hasiotis, 2011*: figs. 5a–5d), and saturated medium sand (*Fairchild & Hasiotis, 2011*: figs. 6e and 6f). SGDS 1290 is preserved in, and was

presumably registered in, a fine-grained sand, lithologically matching one set of experimental conditions in *Fairchild & Hasiotis (2011)*. However, SGDS 1290 is not as detailed as many of the experimentally generated crayfish traces in comparable sediments. This could indicate one or more things: the fossil could be a slight overtrack (sensu *Bertling et al., 2006*) rather than a direct natural cast; the trace maker may have been partly buoyant; and/or trace-maker behavior and sediment consistency combined such that the lighter limbs did not impress as deeply as the heavier tail.

## DISCUSSION

### Trace maker

The stronger resemblance of SGDS 1290 to experimental crayfish locomotion traces than to any known ichnotaxon, or other documented extant arthropod trace, implies a crayfish or crayfish-like maker for the fossil. Whether SGDS 1290 had a crayfish-like or an actual crayfish maker depends on whether the term "crayfish" is used to refer to members of a monophyletic clade (Parastacidae + (Astacidae + (Cambaridae + Cricoidoscelosidae)); *Karasawa, Schweitzer & Feldmann, 2013*) of freshwater (and brackish water if *Protastacus* is included, sensu *Karasawa, Schweitzer & Feldmann, 2013*) lobsters, or, more broadly, to any freshwater, lobster-like crustacean regardless of phylogenetic position, which presumes that more than one crayfish-like lineage colonized terrestrial environments in the past. Here we use the term in the monophyletic sense: true crayfish comprise Astacida (sensu *Karasawa, Schweitzer & Feldmann, 2013*; *Schram & Dixon, 2004*; *Shen, Braband & Scholtz, 2015*). Whether or not the maker of SGDS 1290 was a true crayfish is unclear: the oldest undisputed crayfish body fossils are from the Early Cretaceous (*Garassino, 1997*; *Martin et al., 2008*; *Shen, 2003*; *Taylor, Schram & Shen, 1999*), although unnamed, Late Jurassic specimens from western Colorado also have been called crayfish (*Hasiotis, Kirkland & Callison, 1998*). Despite this, a Triassic or earlier origin for true crayfish has been hypothesized frequently (*Breinholt, Pérez-Losada & Crandall, 2009*; *Crandall & Buhay, 2008*; *Porter, Pérez-Losada & Crandall, 2005*; *Schram, 2001*; *Schram & Dixon, 2004*; *Wolfe et al., 2019*) and possibly substantiated by fossil burrows referred to crayfish from the Early Permian (*Hembree & Swaninger, 2018*) and Late Permian–Early Triassic (*Baucon et al., 2014*; *Hasiotis & Mitchell, 1993*).

Several Late Triassic body-fossil specimens also have been reported as crayfish (*Hasiotis, 1995*; *Hasiotis & Mitchell, 1993*; *Miller & Ash, 1988*; *Olsen & Huber, 1997*; *Santucci & Kirkland, 2010*), but the identities of these specimens as true astacidans has not been established. *Miller & Ash (1988)* placed a Late Triassic specimen from Petrified Forest National Park, Arizona in *Enoploclytia*, which is an erymid lobster, not an astacidan, genus. That generic placement subsequently has been contested (*Amati, Feldmann & Zonneveld, 2004*; *Schweitzer et al., 2010*; *Urreta, 1989*), so the specimen needs detailed restudy, but if it pertains to Erymidae rather than Astacida, then it indicates that a lineage of erymid lobsters colonized terrestrial environments, possibly before true (monophyletic) crayfish. Some older analyses (reviewed in *Rode & Babcock, 2003*) postulated crayfish origins within Erymidae, but more recent phylogenetic analyses (*Devillez, Charbonnier & Barreil, 2019*; *Karasawa, Schweitzer & Feldmann, 2013*;

*Rode & Babcock, 2003*; *Schram & Dixon, 2004*; *Stern & Crandall, 2015*) have recovered (a frequently paraphyletic) Erymidae with members at varying distances from Astacida. If those hypothesized phylogenetic relationships are correct, then no erymid can, in a monophyletic sense, be considered a crayfish, even if it was a freshwater taxon. But regardless of semantics or phylogenetic relationships, crayfish and erymid morphological similarities suggest that their locomotory traces might be indistinguishable, making a definitive attribution of SGDS 1290 impossible.

A crayfish or crayfish-like trace maker for SGDS 1290 is tenable both chronologically and ecologically. As mentioned above, multiple crayfish-like morphotypes have been found in the Upper Triassic Chinle Formation of Arizona and Utah (*Hasiotis, 1995*; *Miller & Ash, 1988*; *Santucci & Kirkland, 2010*). The Moenave Formation overlies the Chinle Formation in southwestern Utah, so crayfish or crayfish-like decapods plausibly could have been present in and around freshwater Lake Whitmore both geographically and stratigraphically. As-yet-undescribed, crayfish or crayfish-like body fossils also have been recovered from lacustrine sediments of the uppermost Triassic Chatham Group (Newark Supergroup) in North Carolina (*Olsen & Huber, 1997*), attesting to how widespread such arthropods were in terrestrial environments in North America even prior to the Jurassic.

### Ichnotaxonomy

To date, no fossil arthropod locomotory ichnotaxon has been attributed definitively to a crayfish or crayfish-like trace maker. Several such ichnotaxa have been attributed, for various reasons, to crustaceans (*Braddy, 2003*; *Gand et al., 2008*; *Lima, Minter & Netto, 2017*; *Pirrie, Feldmann & Buatois, 2004*; *Savage, 1971*; *Walker, Holland & Gardiner, 2003*); additionally, some purported crustacean tracks have not been assigned to particular ichnotaxa (*Imaizumi, 1967*; *Karasawa, Okumura & Naruse, 1990*; *Matsuoka et al., 1993*), including mortichnial trackways leading to *Eryma*, *Eryon*, and *Mecochirus* lobster body fossils from the marine, Upper Jurassic lithographic limestones of Germany (*Glaessner, 1969*: fig. 243A; *Seilacher, 2008*: fig. 2; *Viohl, 1998*: fig. 6). None of these German taxa are crayfish, though morphological similarities of *Eryma* and *Mecochirus* to crayfish might mean that they would have produced indistinguishable locomotory ichnofossils during normal, forward locomotion. None of their traces have been granted detailed description or ichnotaxonomic assignment.

Only three locomotory ichnotaxa have been attributed specifically to crayfish. First, *Heidtke (1990)* erected *Pollichianum repichnum* for Early Permian ichnofossils from Germany that he attributed to the "crawfish" (in the English abstract; "Krebses" in the German abstract) *Uronectes fimbriatus*, also from the Early Permian of Germany. However, *Uronectes* has long been recognized as a syncarid (*Brooks, 1962*; *Calman, 1934*; *Perrier et al., 2006*), not an astacidan, or even a decapod, so the term appearing in the English abstract likely is a simple translation error. Furthermore, however, *P. repichnum* is not differentiable from the resting trace (cubichnion) *P. cubichnum* (*O'Brien, Braddy & Radley, 2009*) and therefore is a junior synonym and is not a locomotory trace. In any case, *Pollichianum* is morphologically quite unlike both SGDS 1290 and experimentally
generated crayfish traces (*Fairchild & Hasiotis, 2011*). Second, *Bolliger & Gubler (1997)* hypothesized that their novel, early Miocene ichnospecies *Hamipes molassicus* was made by a buoyed (presumably swimming) crayfish. *Getty (2018)* referred these specimens to *Conopsoides*; later, *Getty & Burnett (2019)* suggested that at least some of the specimens may pertain to *Acanthichnus*, and they differ from SGDS 1290 for the same reasons outlined above for *Acanthichnus*. Third, *De Gibert et al. (2000)* attributed Early Cretaceous, Spanish specimens that they assigned to *Hamipes didactylus* to crayfish. *Getty (2018)* attributed these tracks to *Bifurculapes* and maintained a crustacean track maker for *H. didactylus*, but was not more specific. However, neither *Bifurculapes* nor *Hamipes* resemble experimentally generated crayfish traces (*Fairchild & Hasiotis, 2011*), or any of the mortichnial decapod traces, and thus are unlikely to have been made by a crayfish-like decapod, at least while walking. Lastly, we also note that unnamed trackways attributed to crayfish from the Upper Triassic Chinle Formation of Utah were mentioned, but not described, by *Hasiotis (1991)*; *Fairchild & Hasiotis (2011)* did not note whether or not these were similar to their experimentally generated traces. Additionally, an unnamed "crayfish locomotion trace" was figured, but not described, by *Rainforth & Lockley (1996*: fig. 1b*)*; it does not resemble either SGDS 1290 or experimentally generated crayfish traces (*Fairchild & Hasiotis, 2011*).

As detailed above, SGDS 1290 does not fit neatly into any existing ichnospecies of *Siskemia*. Whether or not to place it in a novel ichnospecies, or even ichnogenus, is, therefore, an open question. *Bertling et al. (2006)*, *Gaillard (2011b)* and *Minter, Braddy & Davis (2007)* provided solid criteria for the erection of new ichnotaxa, the latter particularly for arthropods. One criterion is that a new ichnotaxon ideally should be represented by a substantial number of specimens that demonstrate behavioral and substrate-based morphological variation; this prevents erecting several ichnotaxa for minor, readily explained variations in trace morphology. SGDS 1290, as a singular specimen, certainly does not meet that criterion, but *Minter, Braddy & Davis (2007)* also allowed that truly unique morphologies exhibited by singular specimens can support an ichnotaxon. In terms of uniqueness, another criterion is whether or not a new morphotype falls onto a continuum, established or hypothetical, of morphologies within an established ichnotaxon. SGDS 1290 is closest morphologically to ichnospecies of *Siskemia*, but has several distinctions from any established ichnospecies therein, particularly the thick and undulating paramedial impressions and the wider spacing between the paramedial impressions and consequent closer appression of the paramedial impressions to the tracks: the average ratios of the distance between a paramedial impression and its flanking track to the distance between the medial edges of the paramedial impressions are 1.34 for *S. bipediculus*, 1.75 for *S. elegans*, and 0.80 for *S. latavia* compared to 0.31 for SGDS 1290 (see Supplemental Material). No published specimen of *Siskemia* demonstrates the features of SGDS 1290; nor do *Siskemia*-like traces made by archaeognathan insects in experimental conditions (*Getty et al., 2013*). Thus, SGDS 1290 does not appear to fall within the established *Siskemia* continuum. The greater prominence (depth) of the paramedial impressions of SGDS 1290 than their associated tracks suggests either a trace maker with heavier tail elements than the gonostyli of an archaeognathan insect or an

archaeognathan trace maker with unusually large styli adopting an unusual posture (possibly partly buoyant), flexing its caudal region downward to create deep styli impressions but not deep track impressions. We consider the latter unlikely; thus, SGDS 1290 does not appear to fall within a hypothetical *Siskemia* continuum, either. However, SGDS 1290 falls within the continuum of trace morphologies made by extant crayfish in experimental conditions (*Fairchild & Hasiotis, 2011*). No philosophical basis has been established for the recognition of novel ichnotaxa on the basis of comparison with traces made by extant organisms; only by comparison with fossil ichnotaxa because extant traces cannot be the basis for an ichnotaxon (*Bertling et al., 2006*; *International Commission on Zoological Nomenclature, 1999*).

SGDS 1290 clearly is morphologically distinctive. Lacking a sufficient number of specimens with which to determine ranges of morphological variation, however, erecting a new ichnogenus for it clearly is unwarranted. Yet we feel that its unique morphology warrants ichnotaxonomic distinction. Given its distant similarity to *Siskemia* ichnospecies, we therefore herein place it in a new ichnospecies of that ichnogenus.

## SYSTEMATIC ICHNOLOGY

**Ichnofamily** Protichnidae *Uchman, Gaździcki & Błażejowski, 2018*

**Ichnogenus** *Siskemia Smith, 1909*

**Type Ichnospecies** *Siskemia elegans Smith, 1909*

**Diagnosis.** Trace consisting of parallel rows of grouped tracks on either side of two parallel, paramedial impressions. Each group of tracks consists of up to four imprints arranged in series, transversely or obliquely to the midline of the trackway (following *Walker, 1985*). *Walker (1985)* further specified that *Siskemia* was diagnosed by paramedial impressions with maximum widths of 0.5 mm, but following *Bertling et al. (2006)*, size is not a suitable ichnotaxobase.

**Ichnospecies** *Siskemia eurypyge* isp. nov.
Figures 3A and 3B

**Diagnosis.** Two parallel, undulating, paramedial impressions flanked externally by closely appressed sets of 1–3 small, ovoid to tapered to elongate tracks; tapered and elongate tracks have long axes parallel or oblique to the trackway axis. Track sets are oriented perpendicular to the trackway axis. Left and right tracks are arranged in a staggered to alternating pattern. Paramedial impressions are mediolaterally thick, but discontinuous, tapering out of existence briefly in some places. Impressions are gently undulating (low amplitude). The paramedial impressions lie far from the trackway axis, generally closer to (and sometimes in contact with) the tracks than to the midline axis or each other.

**Holotype.** Natural cast specimen SGDS 1290, St. George Dinosaur Discovery Site, St. George, Utah, USA.

**Type locality.** "Bug Crossing Quarry," SGDS Loc. 87, St. George Dinosaur Discovery Site, St. George, Washington County, Utah, USA (Fig. 1).

**Stratigraphy.** "Top Surface" of Johnson Farm Sandstone Bed (unit 40 of *Kirkland et al., 2014*), Whitmore Point Member, Moenave Formation (Fig. 2). Hettangian, Lower Jurassic.

**Derivation of name.** From the Greek ευρυς (eurys), meaning "broad" or "wide," and πυγή (pyge), meaning "rump." The ichnospecies name refers to the wider spacing between the paramedial impressions, made by the rear end of the trace maker, than those of other *Siskemia* ichnospecies.

## CONCLUSIONS

SGDS 1290, from the Lower Jurassic (Hettangian) Whitmore Point Member of the Moenave Formation, consists of two paramedial impressions that are flanked by staggered to alternating sets of tapered or elongate tracks. The traces closely resemble those made by extant crayfish (*Fairchild & Hasiotis, 2011*) and are similar in components to traces placed in the ichnogenus *Siskemia* (*Smith, 1909*; *Walker, 1985*). In previously recognized *Siskemia* ichnospecies, the paramedial impressions are thin, relatively straight, and closely appressed to the trackway axis. But in SGDS 1290, paramedial impressions have the opposite morphology and arrangement: they are thick and lie closer to their tracks than the medial axis of the trackway. SGDS 1290 paramedial impressions also undulate, which is not seen in any previously known *Siskemia* ichnospecies. Thus, we erect a new ichnospecies, *Siskemia eurypyge*, to house SGDS 1290 and as-yet undiscovered traces with this morphology and arrangement.

Placing SGDS 1290 in *Siskemia* extends the known range of the ichnogenus into the Early Mesozoic. All other reported occurrences of the ichnogenus are Paleozoic in age: Early Silurian (*McNamara, 2014*; *Trewin & McNamara, 1994*), Late Silurian (*Davies, Sansom & Turner, 2006*), Early Devonian (*Pollard, Steel & Undersrud, 1982*; *Pollard & Walker, 1984*; *Smith, 1909*; *Walker, 1985*), and Pennsylvanian (*Getty et al., 2013*). However, age should not be a factor in ichnotaxonomy (*Bertling et al., 2006*). At least some Paleozoic *Siskemia* traces likely were made by archaeognathan insects (*Getty et al., 2013*), which are extant and for which body fossils are known as early as the Devonian. Based on their similarity to traces made by extant crayfish (*Fairchild & Hasiotis, 2011*), *S. eurypyge* likely was made by a crayfish or crayfish-like crustacean, for which body fossils are known as early as the Late Triassic and which also are extant. Thus, *Siskemia* ispp. traces would be expected to occur from Early Silurian to Recent, but thus far have not been documented except for the occurrences noted above.

SGDS 1290 expands the ichnological record of crayfish and crayfish-like animals to include repichnia in addition to domichnia. Fossil burrows (*Camborygma* ispp.), usually attributed to crayfish, are well known at some sites and in some formations (*Hasiotis, 1995*; *Hasiotis & Bown, 1996*; *Hasiotis & Honey, 1995*; *Hasiotis & Mitchell, 1993*; *Hasiotis, Kirkland & Callison, 1998*; see *Schram & Dixon (2004)* concerning pre-Cretaceous examples), attesting to the presences—and, in some places, abundances—of crayfish and/or crayfish-like taxa in Mesozoic–Cenozoic freshwater paleoecosystems.

Yet locomotion traces made by these burrowers oddly have never before been documented as ichnofossils, possibly because they infrequently venture far from their burrows in substrates suitable for registering locomotory traces, as with modern crayfish (*Martin, 2013*). SGDS 1290 is the first documented locomotory ichnofossil made by a freshwater crayfish or crayfish-like organism, as well as the first fossil evidence of such a taxon in the Lower Jurassic Moenave Formation and indeed the Early Jurassic of the southwestern US. The absence of *Camborygma* burrows in the Moenave Formation that would have been made by the SGDS 1290 trace maker is puzzling, and may be a consequence of a lack of recognition; alternatively, the producer of SGDS 1290 was not a burrower.

## ACKNOWLEDGEMENTS

We thank Patrick R. Getty (Collin College) for invaluable discussion and for loaning us peels of *Siskemia* traces from Scotland. Dianne Aldrich (Dixie State University library) displayed uncanny skill in obtaining copies of often obscure papers via InterLibrary Loan. Kathleen Huber (SEPM) granted permission to reproduce the photo in Fig. 3D from *Fairchild & Hasiotis (2011)*. We thank SGDS volunteer, Jon Cross, for discovering and collecting the specimen. We also thank Patrick Getty, Carlos Neto de Carvalho, and an anonymous reviewer for their time and very helpful comments, which greatly improved the paper.

### Funding

Both the Biology Department and Student Government of Dixie State University provided funding that subsidized this research and its publication. The funders had no role in study design, data collection and analysis, decision to publish, or preparation of the manuscript.

### Grant Disclosures

The following grant information was disclosed by the authors:
Both the Biology Department and Student Government of Dixie State University.

### Competing Interests

The authors declare that they have no competing interests.

### Author Contributions

- Makae Rose conceived and designed the experiments, performed the experiments, analyzed the data, prepared figures and/or tables, authored or reviewed drafts of the paper, and approved the final draft.
- Jerald D. Harris conceived and designed the experiments, performed the experiments, analyzed the data, prepared figures and/or tables, authored or reviewed drafts of the paper, and approved the final draft.

- Andrew R.C. Milner conceived and designed the experiments, performed the experiments, analyzed the data, prepared figures and/or tables, authored or reviewed drafts of the paper, and approved the final draft.

## Data Availability

Specimen SGDS 1290, the holotype of *Siskemia eurypyge* isp. nov., is reposited at the St. George Dinosaur Discovery Site at Johnson Farm (SGDS) in St. George, Washington County, Utah, USA. The specimen is a natural cast from the Bug Crossing Quarry (SGDS locality 87) at the SGDS; stratigraphically, it is from the Top Surface of the Johnson Farm Sandstone Bed within the Whitmore Point Member of the Moenave Formation (age: Hettangian, Early Jurassic).

## New Species Registration

The following information was supplied regarding the registration of a newly described species:

Publication LSID: urn:lsid:zoobank.org:pub:D78963CE-11C8-4447-8E26-BBCCF0E37143.

*Siskemia eurypyge* sp. nov LSID: urn:lsid:zoobank.org:act:769B0815-8991-4F0E-B32C-99C87A9D293B.

## Supplemental Information

Supplemental information for this article can be found online at http://dx.doi.org/10.7717/peerj.10640#supplemental-information.

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
