# Peer review of "A trace fossil made by a walking crayfish or crayfish-like arthropod from the Lower Jurassic Moenave Formation of southwestern Utah, USA"

_PeerJ, doi:10.7717/peerj.10640_

## Round 0.1 · original submission · Major Revisions

The reviews for this manuscript are overall pretty positive, but have enough suggestions to be on that cusp between minor and major revision. Please address the reviewer comments in detail during your revisions. I look forward to seeing the next version of the manuscript!

·

Basic reporting

I think that this is an interesting manuscript that presents unique findings that are of interest to the paleontology community, but it needs revision before it can be published. There are numerous issues with the manuscript as it is written that the authors should address:

The first of these issues is in how the authors use the word ichnotaxon, which is used as a synonym for an ichnospecies. A taxon, however, is any grouping at any rank, so an ichnogenus is also an ichnotaxon. Given that, it is not quite right to claim that the fossil that they describe is not a variant of any given taxon, because its general morphology is the same as Siskemia, as they note on lines 447 through 449. The only objection that could be raised is that Walker (1985) included a width measurement in her diagnosis of Siskemia, but as the present authors note on line 652 (and I agree with this), size is not a valid ichnotaxobase. It is fair to say that their specimen does not fit into any known ichnospecies of Siskemia, but saying that it doesn't fit any taxon is a bit too much of a stretch because its general layout fits the definition of Siskemia sans the measurement.

The second issue is in regard to the ichnotaxonomy and comparison sections, where the authors include track makers in their analysis. See, for example, lines 443 and 444, where they note that Getty et al. (2013) attributed the Siskemia that they studied to archaeognathans. Getty et al. (2013), however, did not imply that all Siskemia are the traces of archaeognathans. Based on the attribution of some specimens to archaeognathans, the present authors reject the idea that their trace falls within the hypothetical continuum for Siskemia. The original Siskemia specimens described by Smith (1909) were probably made by a different group of arthropods, so considering only archaeognathans in a hypothetical continuum is too narrow a view.

Here are some additional comments:

Title- the wording is a bit odd as it is written. Even though the hyphen indicates a compound adjective, I think that most readers will quickly look over the title and come away thinking how does a trace fossil walk. I recommend rewriting it as follows (or something like this): "A trace fossil made by a walking crayfish or crayfish-like arthropod...

Abstract- The whole point of this paper is to describe a previously unknown trace fossil morphotype from the SGDS, so leading off with a statement that the trace fossils from said site are well understood is a bit odd. I think that what the authors mean is that the vertebrate traces are well understood since they have been published on much more, and if this is correct, then I recommend rewording. Also here and elsewhere (as I discuss above), the specimen is a variant of the ichnogenus Siskemia, so it is a variant of a known ichnotaxon.

Introduction- Given that the paper is about a hitherto unknown crayfish or crayfish-like arthropod trace, some attention to what is known about fossil crayfish traces should be mentioned here to set the stage for the paper.

Materials and methods- there is no discussion of which measurements are actually taken. This should be added here, and I also recommend a figure to illustrate these measurements. It's not until we get to the description and the table that we have any idea of what measurements were taken. I also recommend taking measurements of the width between the inside and outside of the track rows and then doing a simple analysis that compares the track row width to the width of the medial impressions between your trackway and those of the other Siskemia species; doing so will show how much your specimen is distinct from previously described specimens.

Description-
Line 140 (and elsewhere in the manuscript)- are the sets closely appressed to each other or are the tracks within the sets closely appressed? It seems like the latter is meant, so rewording should be done to clarify.
Lines 144, 147, and 204- I recommend avoiding anatomical terms for bodies, such as cranial, and stick with words such as anterior.
Line 145- how is staggered with half a cycle of displacement different from alternating?
Line 166- add text about the ratios of width between track rows and width of the medial imprints here

Comparison- this is a very lengthy section for a good reason: the authors conduct an exhaustive comparison between the fossil that they are describing and previously described ichnospecies. I commend them for this because few authors do this. I do think, however, that this section should actually be moved to after the one on the trace maker because most readers will be more interested in what made the trace than what ichnotaxon it belongs to.
Line 212- D aenigma can't be the holotype if it were never designated as such. It could only have lectotype status.
Lines 452-453- the authors could point out here that the medial impressions of other Siskemia ichnospecies don't undulate, whereas the ones of the specimen that they are describing do

Discussion (Trace maker)- this is the most important part of the study, in my opinion. A stronger case should be made here that the trackway is that of a crayfish. The best way to do that is not by eliminating ichnotaxa, but by going to the neoichnological studies. One thing that I think is really important here is a close-up photograph of one of the crayfish trackways illustrated by Fairchild and Hasiotis (2011). A comparison of the trackway from figure 2E of Fairchild and Hasiotis with the fossil trackway would be especially compelling. As it is, the line drawing in Figure 4 (MM to NN) is very small and interpretive. Let the readers see how compelling case is for themselves. If the authors are unable to get an image from Fairchild or Hasiotis, I can provide them with one (although the medial impressions don't undulate in that experimentally produced trackway).

Conclusions- undulating isn't the opposite of linear

References- Olsen and Huber is correct here, but the in-text citation is misspelled as Olson and Huber. Also, Schindewolf (1928 isn't cited in the text).

Figures- I recommend deleting the northeastern USA and the southern tip of Texas from Figure 1A because they look like they are supposed to be part of Utah, but they aren't. I also recommend adding two figures: one on the measurements and another that should be aside-to-side comparison between a modern crayfish trackway and the fossil.

Experimental design

The research background (i.e., what is known of crayfish trace fossils) could be better stated in the introduction.

I also strongly urge some additional measurements be taken. I've already mentioned these above, but I'll expound on them here. One of the three ways that the fossil described in this trackway is different from other ichnospecies of Siskemia is that its medial imprints are widely spaced relative to the rest of the trackway. This difference can be quantified by measuring the outer trackway width and then plotting this relative to the distance between the medial imprints. This will have to be done for the other Siskemia ichnospecies so that the comparison can be made.

Validity of the findings

I think the authors have a strong case that they have found a crayfish or crayfish-like arthropod trackway, but that they could do a better job in showing this to the reader with the side-to-side comparison figure that I recommend.

I do not agree with their statement that the trackway that they describe does not fit into any ichnotaxon, because it fits into the ichnogenus Siskemia. I do suspect, however, that they are right that they have a new ichnospecies, but I'd like them to show some measurements on a figure to demonstrate that the medial imprints are wider relative to the rest of the trackway than they are in other Siskemia ichnospecies.

·

Basic reporting

Clear profissional and native English;
Great literature review; unfortunately it is based on a single specimen
Professional structure, excellent figures
The comparison with crayfish-like trackways is well supported by literature comparison, especially with neoichnological data. Although highly variable morphology of modern crayfish trackways depends on the type and consistence of the substrate the single specimen studied can be positively compared with them. The fact that only up to 3 tracks are found in the trackway' sets of imprints do not discard the possibility of being produced by crayfish but there are insect producers attributed to the ichnogenus Siskemia, to which the specimen was attributed, and also a clearly difference in size. It would be relevant to have different specimens to compare preservational variants. A single specimen do not allow to have definite conclusions.

Experimental design

The paper is based on a rigorous investigation and is basically a ichnotaxonomic paper very well supported by a deep comparison with known ichnotaxa.

Validity of the findings

Until this paper, examples of fossil crayfish trackways have yet to be
described in detail, although they have been presented in figures (e.g.,
Hasiotis, 1991; Rainforth and Lockley, 1996); there were no ichnotaxa
readily attributed to crayfish locomotion.

The fact that only up to 3 tracks are found in each trackway set of imprints do not discard the possibility of being produced by crayfish but there are insect producers attributed to the ichnogenus Siskemia, to which the specimen was attributed, and also a clearly difference in size. It would be relevant to have different specimens to compare preservational variants and locomotory patterns, including the number of legs involved in locomotion. A single specimen do not allow to have definite conclusions.

The single specimen resembles a crayfish-type walking trace and comes from a stratigraphic succession with clear importance for understanding the evolution of early freshwater decapods, Thus, this paper may bring some important implications in the discussion about the evolution of the Astacida.

Additional comments

Great ichnotaxonomical review of arthropod repichnia! I like especially Fig. 4.

---

## Round 0.2 · accepted · Accept

Thank you for your close attention to the reviewer comments. In my view, the manuscript is ready to proceed to the next steps.

·

Basic reporting

I've gone over the manuscript again and think that the authors have made most of the substantive changes that I recommended. I think that the article should be considered ready for publication at this point.

Experimental design

They have addressed my concerns about experimental design

Validity of the findings

the authors have done a better job explaining their findings in the second version of the manuscript